# Multimorbidity clustering of the emergency department patient flow: Impact analysis of new unscheduled care clinics

Adrien Wartelle[1,2]*, Farah Mourad-Chehade[1], Farouk Yalaoui[1], Hélène Questiaux[3], Thomas Monneret[4], Ghislain Soliveau[4], Jan Chrusciel[2], Antoine Duclos[5,6], David Laplanche[2], Stéphane Sanchez[2,7]*

1 Computer Science and Digital Society Laboratory (LIST3N), Université de Technologie de Troyes, Troyes, France, 2 Public Health and Performance Department, Centre Hospitalier de Troyes, Troyes, France, 3 Emergency Department, Centre Hospitalier de Troyes, Troyes, France, 4 Etablissement Aubois des Soins Immédiats, Troyes, France, 5 Research on Healthcare Performance Lab, INSERM U1290 RESHAPE, Université Claude Bernard Lyon 1, Villeurbanne, France, 6 Health Data Department, Hospices Civils de Lyon, Lyon, France, 7 Research on Health University Department–University of Reims Champagne Ardenne, Reims, France

* stephane.sanchez@hcs-sante.fr (SS); adrien.wartelle@ch-troyes.fr (AW)

**Data Availability Statement:** The main dataset containing the patient's exact stay with arrival date, ID code number, stay number, age, gender,

## Abstract

### Background

In France, the number of emergency department (ED) admissions doubled between 1996 and 2016. To cope with the resulting crowding situation, redirecting patients to new health-care services was considered a viable solution which would spread demand more evenly across available healthcare delivery points and render care more efficient. The objective of this study was to analyze the impact of opening new on-demand care services based on variations in patient flow at a large hospital emergency department.

### Methods

We performed a before-and-after study investigating the use of unscheduled care services in the Aube region in eastern France, that focused on ED attendance at Troyes Hospital. A hierarchical clustering based on co-occurrence of diagnoses was applied which divided the population into different multimorbidity profiles. Temporal trends of the resultant clusters were also studied empirically and using regression models. A multivariate logistic regression model was constructed to adjust the periodic effect for appropriate confounders and therefore confirm its presence.

### Results

In total, 120,722 visits to the ED were recorded over a 24-month period (2018–2019) and 16 clusters were identified, accounting for 94.76% of all visits. There was a decrease of 56.77 visits per week in seven specific clusters and an increase of use of unscheduled health care services by 328.12 visits per week.

diagnoses and other information described in the methods section cannot be made publicly available due to patient confidentiality. To avoid any possible patient re-identification and protect anonymity, access to this data is restricted by law to researchers who have complied with all national legislation regarding epidemiological studies. Upon valid request, this data can be obtained from data protection officer of the hospital (Hospital Center of Troyes): dpo@ch-troyes.fr All analyses and data management were performed using R version 3.5.2. The code of the clustering method with intermediate tables for testing is available upon request and requires RStudio software with 3 libraries: Tidyverse, Lubridate and Reshape2.

**Funding:** Our institution received an exceptional funding of 3000 euros for this study from the French Regional Health Agency. The funders had no role in study design, data collection and analysis, decision to publish, or preparation of the manuscript. No authors received salaries from the French Regional Health Agency.

**Competing interests:** The authors have declared that no competing interests exist.

## Conclusions

Using an innovative and reliable methodology to evaluate changes in patient flow through the ED, these findings may help inform public health policy experts on the implementation of unscheduled care services to ease pressure on hospital EDs.

## Introduction

Admissions to emergency departments (EDs) in France doubled between 1996 and 2016, increasing from 10 to 20 million visits annually corresponding to an average growth of 3.5% per year [1]. This led to the crowding and saturation of EDs and has, to date, had negative impact on the quality of care [2] as well as on the working conditions of healthcare professionals (HCPs). Given the complex and systemic nature of this problem, expanding ED capacity and employing more staff are insufficient solutions [3]. Only a detailed study of healthcare needs that includes an analysis of patient flow and patient expectations may enable the development of adequate solutions for healthcare delivery [4–7]. While EDs may reorganize internal procedures to optimize patient flow and delivery of care [8, 9], such changes have not stemmed the constantly accelerating flow of patients entering EDs. Redirecting non-essential ED visits to other healthcare services such as Unscheduled Care Service (UCS) facilities that can treat patients with low acuity conditions as an alternative to EDs [10–15] may therefore be one of the main responses proposed by current healthcare reforms [16, 17].

To evaluate the impact of new UCS on the demand for ED care, it is necessary to quantify patient inflows and to characterize the dynamics of these flows through the ED [18, 19]. Most studies in this field have focused on a global evaluation of avoidable ED visits and the resulting financial savings brought by the lower cost of UCS [10–15]. A segmentation of the complex patient flow patterns obtained by classifying ED visits could help characterize the nature of avoidable patient profiles and thus better understand temporal variations in ED attendance [20].

The analysis of multimorbidity patterns may be an appropriate tool to model the complexity of patients in terms of the diversity in and statistical co-occurrences of various health conditions [21–25]. This type of modeling is usually applied to populations of complex and chronic patients with high multimorbidity however many opportunities to extend this analysis to broader factors like disease pathways in patients are still possible [26, 27].

Segmenting the overall ED patient flow using diagnoses from ED visits as well as disease clusters could be an innovative and original approach to evaluate the structure and trends of ED patient flow [22, 28, 29]. We hypothesized that modeling ED patient profiles could provide a more detailed analysis of the impact of UCS on ED attendance. The objective of this study was to investigate the impact of opening new UCS based on patient flows in the largest ED of the Aube Department in France.

## Materials and methods

### Study design and population

A before-and-after study of the use of emergency care services at the Troyes Hospital in eastern France was performed. Troyes Hospital is the largest hospital in the Aube Department of France which has a population of 310,000 inhabitants and a medical density of 234.1 physicians per 100,000 inhabitants. This region medical density in the lowest quartile of all the regions (referred to as Departments in French) in France. The hospital has 442 medical beds, 127 surgical beds and 63 beds dedicated to gynecology/obstetrics. Between October and

November 2018 the opening of two new UCS facilities offering services akin to convenient care clinics changed the organization of emergency care delivery in the city [30]. These two new structures are the *Aubois Establishment of Immediate Care* situated in the north of the city at about 4 km from the hospital and *the Polyclinic of Montier La Celle* at about 1.5km. They are as easy to access as the ED for the surrounding population and thus their localization does not cause any flow constraint. The three structures represent the main unscheduled services in the city and in the department with only one other ED in the Aube department in Romilly sur-Seine accounting for approximately 20,000 ED visits in 2017. They are completed by the Ambulatory unscheduled care, performed for the most part by general practitioners and nurses in the Aube territory via the *SOS Médecin* association and the *Permanence Des Soins Ambulatoires* (PDSA). The Hospital of Troyes entails almost all the hospitalization health capacity of the city and is a necessary passage point for all inpatients requiring emergent care. More information about the health capacity of the city can be found on the FINESS file freely available on internet, or through the hospital website [31].

The study population included the entire population of patients receiving care in the ED of the hospital from 1 May 2017 to 28 April 2019. In 2018, there were a total of 62,082 ED visits corresponding to an average use rate of 250 to 330 visits per 1,000 inhabitants within the hospital's service area. With >45,000 annual ED visits, the ED has been classified as having a very high volume of activity according to national statistics [32]. The 2-year study period was chosen to account for the yearly seasonal variations and to avoid being impacted by a national restructuring of the country's ED circuit that was introduced in 2016 [9].

## Measured variables

The primary endpoint was the difference in patient flow (measured in visits per week) both before and after the opening of two new UCS services by level of subgroups identified using clustering methods.

Three major data points were recorded: i) information about patient stay, visit identifier code, patient identifier code, date of arrival at the ED in addition to subsequent wards and the triage path followed at patient arrival. A triage path was either a long path with normalized evaluation and care management, or a short (fast-track) path which provided a pathway for assessment and treatment of low-severity patients with mostly minor injuries or benign medical conditions. This triage process is done by a triage nurse which evaluates the urgency of the patient through the CIMU scale (Nurse Classification of ED Patients) [33]. This scale has five levels (1, 2, 3, 4 and 5) corresponding in the current context to a necessity of an immediate intervention, of an intervention below 20 mins, 60 mins, 120 mins and 240 mins respectively. The triage evaluation is directly related to the decision sent to the patient to short circuit, long circuit, or vital emergency room for reanimation. During the study period, levels 1, 2, 3, 4 and 5 patients were sent in long circuit or vital emergency for 92.03% of stays (63.2% being vital), 85.55% (3.5% vital), 62.30% (0.54% vital), 15,26% (0.13% vital) and 6.3% (0.08% vital) respectively. We recorded discharge destinations from the ED, patient statuses using the Patient State (PS) classification (previously developed by our group) [9] indicating patient severity. Most patients are assessed with i) one diagnosis (rarely two) that was recorded according to the International Classification of Diseases 10th revision (ICD10), this assessment is made at the end of each stay by the current emergency physicians and resident medical staff with evaluation of all the information given during a patient stay in the ED and is further verified by the medical information team for coherence, ii) information about prescriptions including medication in addition to biological and radiological examination results and iii) socio-demographic data on each patient including age, sex and area of residence (town or postcode).

## Statistical analysis

A multimorbidity clustering method was developed and applied to determine the spectrum of patient profiles according to clinical characteristics. This method was chosen because it revealed more relevant groups to analyze than the PS classification system, ICD chapters, other more traditional clustering approaches (such as k-means, k-mode) and the principal component analyses, which are not well adapted to the categorical nature of patient data due to being unable to provide any meaningful clustering separation. Moreover, the nature of the health problem, revealed by multimorbidity cluster studies seemed to be an excellent proxy for characterizing the clinical profiles of the patients.

A detailed study of the clustering method, concepts and results has been published elsewhere and is summarized below [34]. Using the co-occurrences of ICD10 diagnoses with block level representation, we applied a new design of Hierarchical Agglomerative Clustering (HAC) [35] with a new measure of similarity which targets relative risk:

$$RR_{ij} = \frac{p_{ij}}{p_i p_j} \tag{1}$$

where $p_{ij}$ designates the probability of diagnoses $i$ and $j$ co-occurring in sets of patients, 'visits less than 6 months focus on more meaningful co-occurrences links', whereas, $p_i$ and $p_j$ are the marginal probabilities of occurrences used to weight the relation [36]. Here, the number of clusters was determined by the maximization of a membership ratio (MR) criterion:

$$MR = \frac{RR_{intra}}{RR_{inter}} \tag{2}$$

where $RR_{intra}$ designates the average of the relative risk of a block with the other blocks in its cluster, and $RR_{inter}$ designates the maximum average of the relative risk of a block with a cluster other than its own. Using this criterion, clusters were obtained and arranged by the size of the patient population concerned, then named based on an analysis of the content of the blocks. Relevant information was summarized using means and standard deviations, median and quartiles in addition to numbers and percentages. The content of the blocks of diagnoses and the values of $RR_{intra}$ and $RR_{inter}$ for each cluster were also analyzed.

Differential analysis was performed on the rates of weekly ED visits before and after the opening of the UCS to identify clusters showing a statically significant decrease by using the Fisher's F test for linear trends (statically significance of tests was considered for p-value below 0.05 in this study). The clusters showing a significant decrease were then grouped and decomposed using the CIMU scale to perform another differential analysis and show which acuity of patient was the most impacted. Even though the distribution of the number of weekly visits is Poisson by nature, at a large enough rate this distribution can be well approximated using a Gaussian distribution. A classic linear approach with Gaussian errors is pertinent to judge the trends of mean arrival rates on large scale time periods. Furthermore, using 18 months for the duration of the before period and six months for the after period ensured that the observed trend was strong enough and not just the result of a seasonal variation. In addition to the control for typical seasonal trends, the duration of the after periods with its 1 to 3 ratio, was designed such that we could quantify the medium-term impact where the variations in ED visits had more chance to be directly related to the opening of the new structures than in the long-term case where other complex effects and trends, such as the COVID-19 pandemic, could take place and mitigate the result significance through confounding.

To adjust this for seasonal confounders and obtain confidence interval, the differential analysis was complemented using a time series analysis of the ED visits from the year 2016 to 2019.

Each cluster daily visit count $V_{ED}(t)$ ($t$: day index) over the 4-year period was analyzed with a Poisson distributed linear regression with the retained model inspired by the literature [37, 38]:

$$V_{ED}(t) = a_0 + m(t)'a_1 + d(t)'a_2 + p(t)'a_3$$

where $a_0$ is the intercept, $a_1$ to $a_3$ the regressed linear vector coefficients on the month boolean indicator vector $m(t)$, the day of week boolean indicator vector $d(t)$ and the period boolean indicator vector $p(t)$ with 2 before periods 2016-01-01 to 2017-05-01 and 2017-05-01 to 2018-10-08 (taken as reference) and 2 after periods 2018-10-08 to 2019-04-28 and 2019-04-28 to 2019-12-31. Another model was also applied to consider a global trend effect $a_4$ over $t$.

A multivariate logistic regression model of the probability of a visit belonging to a cluster on the decline was constructed and adjusted for confounders. This model was used to confirm the period effect observed by the differential analysis and to compare the decreasing clusters with the other ones. The final model was selected using a backward selection approach based on minimization of the Akaike Information Criterion (AIC) [39]. The equation of the retained model was:

$$\text{Logit}(p) = \alpha + \omega Pe + \gamma R + V'\rho + X'\beta \tag{3}$$

where p is the probability of belonging and alpha is the constant. Pe is the period (before or after), R indicates whether the visit was followed by readmission within seven days. V is a visit-related informational vector which includes the triage circuit and the quantity of drug prescriptions and exams and X is a patient-related vector that includes the age category, sex, and patient status at arrival to the ED.

In parallel, data regarding unscheduled ambulatory care in the region was recorded based on statistics from the National Health Information database. Overall trends in patient flow were studied from 2016 to 2019 and put into perspective with the results of our analyses.

## Ethics statement

This study was performed in compliance with the national legislation regarding epidemiological studies (Declaration N˚ 2203674v0 on 24 July 2018). Since the study was wholly observational and only used anonymized data (patient names were not recorded), neither ethics approval nor a specific written informed consent from participants were required under French law as a retrospective database study. Moreover, in accordance with national ethical directives, the requirement for written informed consent was waived because the study was strictly observational and all data were blinded. (Ref: French Public Health Code. Article R. 1121–2. [http://www.legifrance.gouv.fr]). According to the French Public Health Code, this research also did not need require an ethical committee (Ref: French Public Health Code. Article R. 1121–2. [http://www.legifrance.gouv.fr]). The study was conducted with all medical information related legislation and was declared to the National Registry of Health research body under N˚1113130319. Patients were informed that the study was being carried out via the hospital's registry of ongoing studies.

## Software

The data used in this study was extracted using from the software's database of the ED of Troyes, named RESURGENCES, and using SQL request through the SQL developer software. This software is directly used by the physicians and residents of the ED to register patients' data during their stay. The data was analyzed using R 4.0.2 in the RStudio 1.2.5042 environment with the packages Tidyverse 1.3.0, Lubridate 1.7.9, Stats 4.0.2, and Treemap 2.4.2.

## Results

During the 2-year study period, 120,722 visits to the ED were recorded involving 75,283 patients. In total, 114,391 of these visits (94.76%) were included in the analysis (Fig 1). These involved 72,666 patients (96.52%), a total of 151 blocks of diagnoses, 2,750 complete ICD10 diagnostic codes and an average of 1.47±0.96 diagnostic occurrences per patient. A total of 6,123 visits (5.07%) involving 5,455 patients (7.26%) were excluded from the clustering analysis due to missing ICD10 diagnostic codes. A further 208 visits (0.17%) were rejected because of an insufficient number of occurrences of the block (defined as a minimum of 10 for the objective of this study).

Among the 114,391 ED visits, where the mean age was 39.84 years (with a standard deviation of 27.40 years), there were 50.86% females, and a mean number of 1.57, visits per patient. The mean number of medications per visit was 1.45 (with a standard deviation of 2.87), the probability of biological exams was 42.9%, and the probability of radiological exams was 31.7%. The ratio of short to long path was 1.81 and the hospitalization rate was 22%.

The cluster analysis aggregated the 114,391 visits into 16 clusters. As illustrated in Table 1 and Fig 2, the populations of these clusters ranged from 1,090 to 13,186 patients and from 1,139 to 17,614 visits with the first five clusters accounting for 72,108 (63.0%) of total visits (see also S1 Table and Fig 2). With the classification of multimorbidity patterns, the clustering allowed for a better assessment of the ED population characteristics.

As described in the original clustering study, Cluster 1 contained a majority of women (71%), and Cluster 2, 4 and 8 encompassed the most care-intensive and elderly populations (with a mean age between 49.58 and 55.53). Cluster 3 contained the youngest population with 51.2% of the total population of children (5 years old or younger) and was mostly related to winter epidemic outbreaks with 23.4% of visits from December and January (instead of the expected 16.67%). Lastly, Clusters 6 to 16 contained most men with proportions up to 64% and seven clusters (Cluster 6, 7, 8, 9, 12, 15 and 16) were directly related to trauma problems,

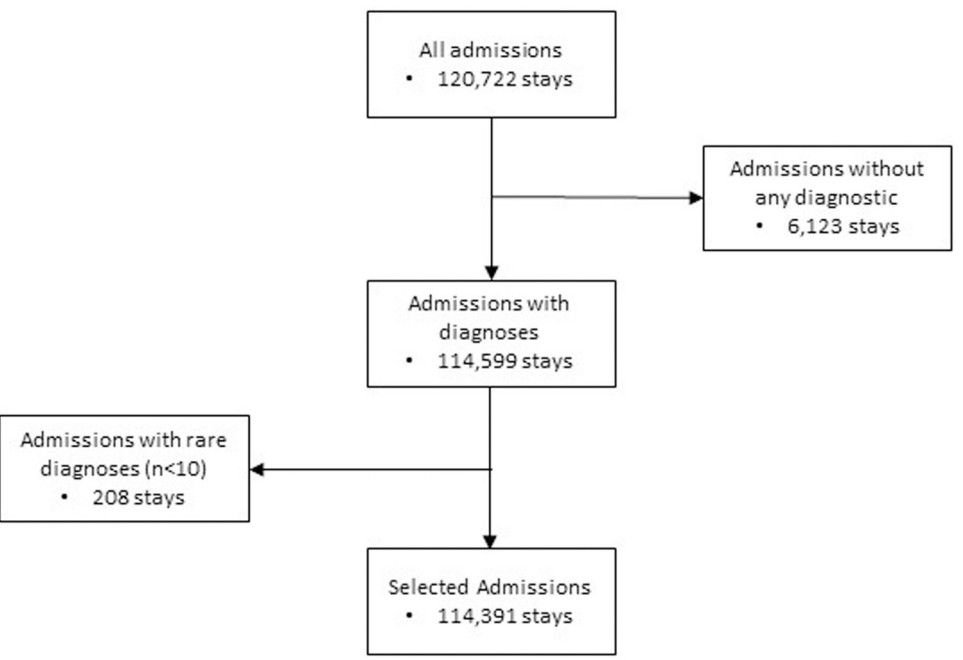

**Fig 1. Visit selection flow chart from the RESURGENCES database.**

**Table 1. Characteristics of the 16 clusters to assess the impact analysis of new unscheduled care services (UCS) in clinics.**

| Cluster name and number | 1 Digestive disorders, pregnancy, menstruation | 2 General symptoms and mental disorders | 3 Infectious diseases | 4 General symptoms of chronic conditions | 5 Mental disorders and at-risk behaviors | 6 Wrist and Hand Trauma | 7 Head Trauma | 8 Hip related trauma and disorders |
|---|---|---|---|---|---|---|---|---|
| Population, n(%) | 13,186 (18%) | 12,195 (17%) | 11,148 (15%) | 11,109 (15%) | 9,541 (13%) | 6,296 (9%) | 5,731 (8%) | 4,866 (7%) |
| Age, mean±SD | 36.47 (±22.88) | 49.53 (±28.86) | 20.76 (±24.79) | 57.81 (±26.36) | 38.61 (±22.56) | 32.23 (±20.96) | 32.23 (±30.24) | 55.25 (±29.84) |
| CIMU, mean±SD | 3.59 (+-0.77) | 3.48 (+-0.93) | 4.01 (+-0.94) | 3.05 (+-0.88) | 3.83 (+-0.88) | 4.30 (+-0.72) | 4.01 (+-0.78) | 3.61 (+-0.91) |
| Females, n(%) | 12,496 (71%) | 7,551 (53%) | 7,040 (49%) | 6,772 (49%) | 6,081 (50%) | 2,599 (38%) | 2357 (39%) | 2,288 (40%) |
| Visits, n(%) | 17,614 (15%) | 14,237 (12%) | 14,271 (12%) | 13,788 (12%) | 12,083 (11%) | 6,759 (6%) | 6087 (5%) | 5,731 (5%) |
| Number of visits per patient | 1.34 | 1.17 | 1.28 | 1.24 | 1.27 | 1.07 | 1.06 | 1.18 |
| Waiting time, minutes; median (Q1-Q3) | 60.12 (29.84–112.00) | 57.57 (27.21–117.65) | 43.82 (22.97–77.11) | 45.28 (20.38–96.83) | 44.13 (18.12–93.95) | 58.80 (30.88–105.18) | 57.23 (27.88–104.31) | 54.58 (26.28–106.08) |
| Duration of management, minutes; median (Q1-Q3) | 166.77 (49.31–319.53) | 233.76 (101.94–379.03) | 64.70 (31.25–153.85) | 277.30 (163.65–419.50) | 137.48 (61.23–281.17) | 70.63 (43.42–113.75) | 69.68 (33.53–167.26) | 184.44 (80.58–322.85) |
| Number of medications, mean ±SD | 1.75 (±2.77) | 2.07 (±3.25) | 0.96 (±2.30) | 2.95 (±4.20) | 0.90 (±2.04) | 0.35 (±0.96) | 0.56 (±1.71) | 2.45 (±3.78) |
| Number of biology exams, mean±SD | 1.98 (±1.96) | 2.15 (±2.17) | 0.73 (±1.49) | 3.06 (±2.21) | 1.19 (±1.91) | 0.11 (±0.49) | 0.30 (±1.00) | 1.68 (±1.91) |
| Number of radiological exams, mean±SD | 1.26 (±1.97) | 1.85 (+-2.20) | 0.50 (±1.35) | 2.84 (±2.30) | 0.95 (±1.80) | 0.10 (±0.46) | 0.29 (±0.99) | 1.25 (±1.92) |
| Number of blocks of diagnoses | 19 | 8 | 20 | 33 | 15 | 1 | 1 | 19 |
| Number of ICD10 diagnostic codes | 333 | 157 | 284 | 354 | 330 | 52 | 68 | 302 |
| Ratio of short circuit to long circuit | 1.18 | 0.61 | 3.97 | 0.24 | 1.97 | 135.18 | 9.46 | 1.13 |
| Patients in PS1 status | 13,552 (77%) | 9,213 (65%) | 12,421 (87%) | 7,336 (53%) | 8,713 (72%) | 6,298 (93%) | 5,488 (90%) | 3,449 (60%) |
| Number admitted to hospital, n(%) | 3,751 (21%) | 4,759 (33%) | 1,756 (12%) | 6,093 (44%) | 2,677 (22%) | 368 (5%) | 519 (9%) | 2,095 (37%) |
| Rate of readmission within 7 days (%) | 9.36% | 5.49% | 6.47% | 4.84% | 6.71% | 3.88% | 3.61% | 7.17% |
| Rate of readmission between 7 and 30 days (%) | 7.94% | 5.98% | 6.13% | 7.72% | 7.22% | 3.03% | 3.48% | 7.31% |
| Cluster number and name | 9 Feet Trauma | 10 Back and Spine disorders | 11 Oculomotor disorders | 12 Lower limb trauma | 14 Arthropathies | 13 Cutaneous infections, wounds, and skin disorders | 15 Shoulder and arm trauma | 16 Chest trauma and other diseases of the pleura |
| Population, n(%) | 4,822 (7%) | 3,455 (5%) | 3,036 (4%) | 2,919 (4%) | 2,601 (4%) | 2,374 (3%) | 1,822 (3%) | 1,090 (2%) |
| Age, mean±SD | 29.81 (±19.56) | 45.44 (±22.51) | 39.42 (±23.86) | 36.12 (±23.45) | 45.78 (±25.36) | 34.73 (±21.09) | 40.91 (±26.85) | 48.80 (±23.27) |
| CIMU, mean±SD | 4.59 (+-0.61) | 4.02 (+-0.87) | 4.15 (+-0.80) | 4.40 (+-0.71) | 4.19 (+-0.84) | 4.44 (+-0.72) | 4.19 (+-0.86) | 3.67 (+-0.95) |
| Females, n(%) | 2,493 (48%) | 2,056 (55%) | 1,256 (39%) | 1,319 (43%) | 1,304 (48%) | 1,326 (48%) | 834 (43%) | 412 (36%) |

*(Continued)*

**Table 1.** (Continued)

| | | | | | | | | |
|---|---|---|---|---|---|---|---|---|
| **Visits, n(%)** | 5,209 (5%) | 3,756 (3%) | 3,224 (3%) | 3,060 (3%) | 2,737 (2%) | 2,777 (2%) | 1,919 (2%) | 1,139 (1%) |
| **Number of visits per patient** | 1.08 | 1.09 | 1.06 | 1.05 | 1.05 | 1.17 | 1.05 | 1.04 |
| **Waiting time, minutes; median (Q1-Q3)** | 61.75 (32.84–105.60) | 62.00 (32.02–111.50) | 55.22 (29.32–94.69) | 61.33 (31.62–109.35) | 60.44 (32.47–110.64) | 54.89 (28.80–95.97) | 48.42 (21.90–92.75) | 56.38 (28.38–101.92) |
| **Duration of management, minutes; median (Q1-Q3)** | 71.03 (44.98–109.19) | 125.00 (55.37–275.67) | 44.10 (23.27–90.77) | 87.90 (52.35–142.57) | 95.04 (44.78–222.75) | 57.33 (26.99–120.95) | 104.36 (61.59–177.55) | 139.88 (68.60–305.92) |
| **Number of medications, mean ±SD** | 0.25 (±1.01) | 1.77 (±2.97) | 0.44 (±1.28) | 0.62 (±1.72) | 1.21 (±2.64) | 0.77 (±1.85) | 1.08 (±2.20) | 1.68 (±2.99) |
| **Number of biology exams, mean±SD** | 0.08 (±0.44) | 0.76 (±1.34) | 0.20 (±0.92) | 0.26 (±0.81) | 0.96 (±1.85) | 0.63 (±1.43) | 0.40 (±0.99) | 1.11 (±1.78) |
| **Number of radiological exams, mean±SD** | 0.07 (±0.43) | 0.60 (±1.27) | 0.16 (±0.81) | 0.24 (±0.79) | 0.85 (±1.79) | 0.31 (±1.21) | 0.39 (±0.98) | 1.09 (±1.77) |
| **Number of blocks of diagnoses** | 2 | 2 | 11 | 1 | 2 | 14 | 1 | 2 |
| **Number of ICD10 diagnostic codes** | 76 | 118 | 94 | 34 | 304 | 187 | 30 | 27 |
| **Ratio of short circuit to long circuit** | 66.64 | 2.89 | 14.11 | 36.22 | 3.59 | 9.16 | 23.14 | 2.21 |
| **Patients in PS1 status** | 5,038 (97%) | 3,032 (81%) | 3,048 (95%) | 2,720 (89%) | 2,245 (82%) | 2,337 (84%) | 1,593 (83%) | 812 (71%) |
| **Number admitted to hospital, n(%)** | 136 (3%) | 672 (18%) | 151 (5%) | 320 (10%) | 466 (17%) | 412 (15%) | 302 (16%) | 297 (26%) |
| **Rate of readmission within 7 days (%)** | 3.11% | 4.77% | 3.85% | 4.18% | 4.57% | 10.08% | 4.48% | 3.51% |
| **Rate of readmission between 7 and 30 days (%)** | 3.63% | 5.01% | 3.01% | 3.43% | 4.68% | 7.31% | 3.39% | 4.04% |

while the other four (Cluster 10, 11, 13 and 14) were indirectly linked to trauma cases. This is associated with 28.61% of the ED visits resulting in diagnoses from Chapter XIX (S00T88): *Injury, poisoning and certain other consequences of external causes.*

The before-and-after analysis showed an overall decrease of 32.48 visits per week for the 16 clusters (2.93%, F-test, p = 0.0355) and a specific decrease of 55.81 visits per week in 7 clusters (13.76%, F-test, p<0.001). Clusters 2, 6, 7, 9, 11, 12 and 15 were associated with this decrease and showed downward trends (as illustrated by Fig 3, S1 Fig, and described in Table 2). The specific decrease corresponded to a loss of 0.083 visits per patient over a period of 2 years. The before-and-after analysis also showed a significant (F-test, p = 0.00251) increase of 18.99 visits per week (14.78%) in Cluster 4 as well as a non-significant (F-test, p = 0.48) increase of 17.56 visits per week (13.19%) for Cluster 3. As illustrated in Table 3, further analysis using the CIMU scale show that only CIMU levels 4 and 5 patients are responsible for a significant decrease of 56.77 visits per week. The Poisson regression model coefficients results available in S3 Table showed consistent but smaller results with an overall decrease of 39.83 visits per week (95% CI [24.49,55.13]) and a specific decrease in the seven clusters of 39.17 visits (95% CI [30.25, 48.06]). The more pronounced overall decrease could be linked to the accounting of

Count of visits per cluster and block of diagnoses

**Fig 2. Distribution of blocks of diagnoses in the 14 clusters.** Each cluster is represented by a unique color. The volume of each area is proportional to the number of patients affected by the given block of diagnoses during the study period. The label of each block can be found at: https://www.icd10data.com/ICD10CM/Codes or in S2 Table.

seasonal effects in Cluster 3 and Cluster 4 where the decrease of 4.09 (95% CI [-1.50,9.63]) and the increase of 3.30 (95% CI [-2.19,8.83]) were much smaller and not significant. The trends of the initial decreasing Clusters 2, 6, 7, 9, 11, 12 and 15 were 8.40 (95% CI [3.03,13.74]), 8.39 (95% CI [4.76,11.98]), 6.21 (95% CI [2.71,9.67]) 8.00 (95% CI [11.78,17.17]), 3.41 (95% CI [0.84,5.94]), 2.51 (95% CI [0.03,5.01]) and 1.70 (95% CI [0.41,3.78]) respectively. They tended to be lower than the empirical observations and Cluster 12 and 15 were not significant anymore. The Poisson regression model with global trend described in S4 Table suggest a more pronounced overall decrease of 85.37 (95% CI [65.58, 105.16]) and specific decrease in Clusters 2, 6, 7, 9, 11, 12 and 15 of 48.99 (95% CI [37.33,60.63]). Using this model that described a more global effect, we would have selected Cluster 1, 2, 3, 6, 7, 9 and 11 that were decreasing significantly.

For the structural analysis of undiagnosed ED visits, there was a non-significant increase of 2.37 visits per week (F-test, p = 0.64). This rose from an average of 58.25 visits per week in the before period to 60.57 in the after period. More than two-thirds (4269 visits, 69.72%) of these visits involved patients who left the ED without waiting for a medical examination. The rest of the patients (1854 visits, 30.28%) could not be diagnosed due to lack of information or due to their nature of stay.

The average number of ED visits decreased linearly by 29.37 visits per week (2.5%) from 1168.66±76.62 weekly visits before the opening of the USC to 1139.29±51.83 visits per week afterward (F-test, p = 0.00000161).

Unscheduled care in the Aube department increased by 328.12 visits per week (14.75%, F-test, p = 0.000425) during the study period. Additional analysis of the National Healthcare Information database illustrated in Fig 4 showed that during the first six months after the opening of the UCS, unscheduled care went increased from 2225.13±258.43 to 2553.25 ±261.27 visits per week. This increase was mainly due to the opening of UCS and accounted

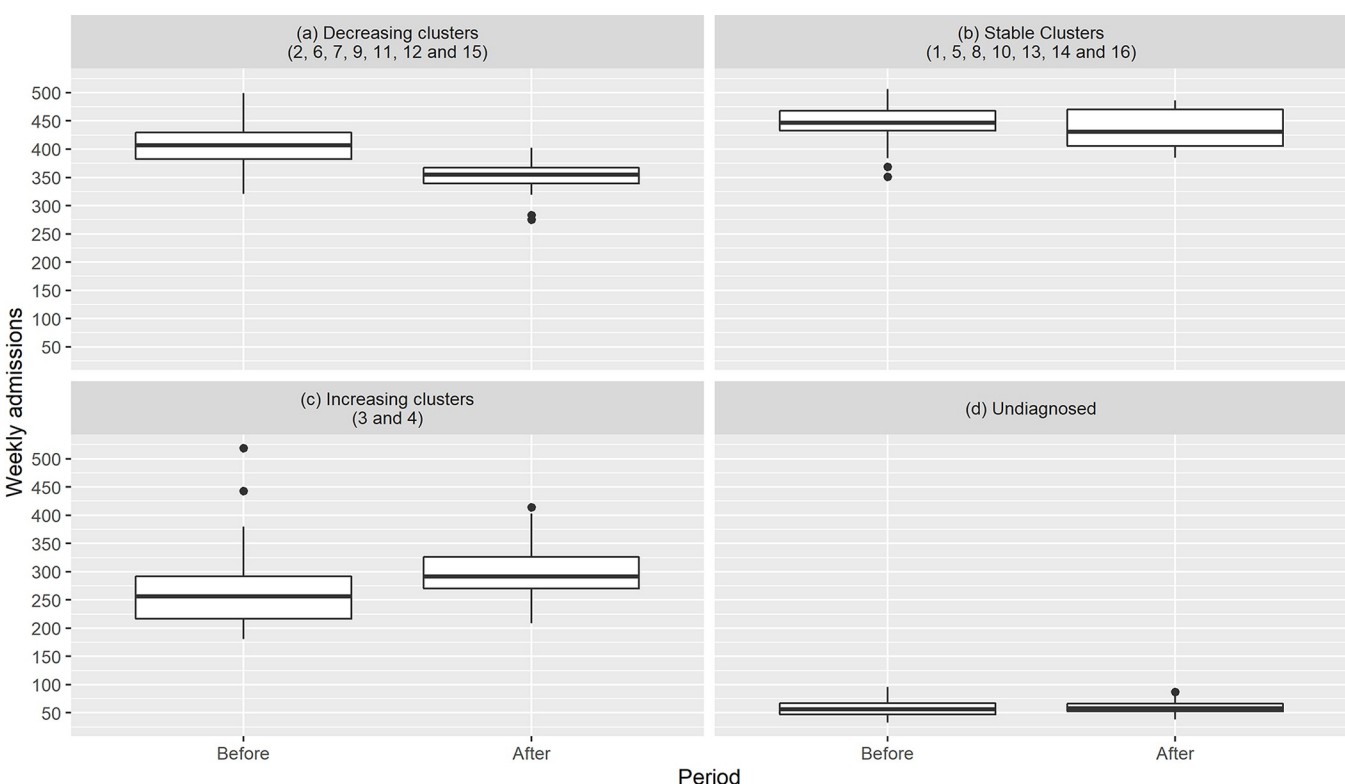

**Fig 3. Trends in clusters before and after the opening of UCS.** The numbers of weekly admissions are represented with boxplots where the middle line indicates the median and the boxes delimit the quartiles. The upper and lower whisker extends from the hinge to the largest and smallest (resp.) value no further than 1.5 * IQR from the hinge (where IQR is the inter-quartile range, or distance between the first and third quartiles).

for 189.5±62.54 visits per week. Using this volume as a reference, the ratio of the specific cluster decrease observed in the ED to the increase seen in UCS was 30.58%.

In the multivariate analysis, the probability of belonging to one of the seven clusters that exhibited significant reductions decreased after the opening of UCS. As illustrated by Table 4, a decrease in these clusters corresponded with the opening of UCS, with odds ratio (OR) of 0.83 (95%CI [0.80 to 0.85]). The ORs of the different variables indicated that the concerned population from the seven clusters had less weekly readmissions (OR 0.58, (95%CI [0.55 to 0.61]), and was more often male with an OR of 0.72 (95%CI [0.70 to 0.74]). This population also had a more moderate disease state with an OR of 1.14 (95%CI [1.09 to 1.18]) for PS1 status, was older with an OR of 1.82 (95%CI [1.74 to 1.90]) for those aged 75 years and older, had a slightly lower need for drugs with an OR of 0.97 (95%CI [0.96 to 0.97]), a lower need of biological examinations with an OR of 0.68 (95%CI [0.66 to 0.69]) and a greater need for radiological examinations with an OR of 1.36 (95%CI [1.33 to 1.39]).

## Discussion

Our results showed a decrease in the probability of belonging to seven clusters after the opening of UCS, that is the recourses for the type of health problem entailed by these clusters has decreased in proportion. The decrease predominantly involved patient profiles linked to trauma and trauma-related symptoms that are low-acuity, meaning they are strictly related to CIMU levels 4 and 5 with patients going predominantly to short circuit. This decline corresponded with the type of patients managed by the UCS in question. Our findings further

**Table 2. Before-after analysis of the weekly visits for each cluster to assess the impact analysis of new unscheduled care services (UCS) in clinics.**

| Cluster number and name | Before UCS opening | After UCS opening | Difference | p-value |
|---|---|---|---|---|
| *1 Digestive disorders, pregnancy, menstruation* | 171.49 ± 13.91 | 166.86 ± 15.06 | -4.63 (-2.70%) | 0.474 |
| *2 General symptoms and mental disorders* | 140.99 ± 14.84 | 129.32 ± 15.06 | **-11.67 (-8.27%)** | **<0.001** |
| *3 Infectious diseases* | 133.12 ± 39.93 | 150.68 ± 31.75 | 17.56 (13.19%) | 0.482 |
| *4 General symptoms of chronic conditions* | 128.47 ± 22.32 | 147.46 ± 24.68 | **18.99 (14.78%)** | **0.00251** |
| *5 Mental disorders and at-risk behaviors* | 118.41 ± 17.48 | 112.14 ± 16.37 | -6.27 (-5.29%) | 0.237 |
| *6 Wrist and Hand Trauma* | 68.04 ± 11.25 | 57.36 ± 8.68 | **-10.68 (-15.70%)** | **0.00302** |
| *7 Head Trauma* | 60.67 ± 9.08 | 53.61 ± 8.35 | **-7.06 (-11.64%)** | **0.00219** |
| *8 Hip related trauma and disorders* | 56.55 ± 8.50 | 52.96 ± 7.88 | -3.59 (-6.35%) | 0.363 |
| *9 Feet Trauma* | 53.97 ± 11.66 | 40.07 ± 7.27 | **-13.90 (-25.76%)** | **<0.001** |
| *10 Back and Spine disorders* | 35.79 ± 6.65 | 37.75 ± 5.34 | 1.96 (5.48%) | 0.166 |
| *11 Oculomotor disorders* | 32.25 ± 6.00 | 28.00 ± 5.93 | **-4.25 (-13.18%)** | **0.0129** |
| *12 Lower limb trauma* | 30.95 ± 7.95 | 25.79 ± 6.47 | **-5.16 (-16.68%)** | **0.00158** |
| *14 Arthropathies* | 26.95 ± 5.64 | 25.43 ± 6.13 | -1.52 (-5.64%) | 0.885 |
| *13 Cutaneous infections, wounds, and skin disorders* | 26.78 ± 6.08 | 26.86 ± 4.58 | 0.08 (0.30%) | 0.734 |
| *15 Shoulder and arm trauma* | 19.47 ± 4.45 | 16.07 ± 4.58 | **-3.40 (-17.47%)** | **0.0046** |
| *16 Chest trauma and other diseases of the pleura* | 10.89 ± 3.36 | 11.54 ± 3.39 | 0.64 (5.88%) | 0.876 |
| **Total** | 1108.66 ± 72.71 | 1076.18 ± 49.76 | **-32.48 (-2.93%)** | **0.0355** |
| **Total for negative trends** | 405.67 ± 34.13 | 349.86 ± 27.34 | **-55.81 (-13.76%)** | **<0.001** |

Note: For each series, the linear trend of the 2-year study period has been tested using Fisher's F test and resulting in a p-value. UCS; Unscheduled Care Services, SD; Standard deviation, *Fisher test on linear trend (F test).

showed a non-significant increase in infectious diseases and a significant increase in general symptoms of chronic conditions. These increases partially compensated for the decrease in visits of the seven clusters. As illustrated in S1 Fig and shown by the Poisson regression model in S3 Table, the temporal variations in the patient profiles of the two clusters are only due to seasonal variations notably during the winter periods of epidemics and crowding of the ED that are more present in the after period, rather than a tangible trend related to the opening of UCS. Consequently, the implementation of UCS was associated with a change in patient flow to the ED. However, the more pronounced visit increase through the UCS, which was of greater magnitude than the reduction in ED visits, suggests that UCS may also have diverted the flow of patients that would have been otherwise treated by appointment-based care. The Poisson regression model with global trend described in S4 Table also suggests that the UCS disrupted the linear trend of ED visits growth that were taking place before 2019 even though the decrease observed is still inferior and the estimation and interpretation of a global linear trend is questionable.

**Table 3. Before-after analysis of the weekly visits for each CIMU level in the *decreasing clusters* to assess the impact analysis of new unscheduled care services (UCSs) in clinics on acuity.**

| CIMU level | Before UCS opening | After UCS opening | Difference | p-value |
|---|---|---|---|---|
| 1 | 1.40 ± 0.64 | 1.59 ± 1.06 | 0.19 (13.78%) | 0.890 |
| 2 | 18.96 ± 4.80 | 20.68 ± 5.08 | **1.72 (9.06%)** | **0.0234** |
| 3 | 99.20 ± 12.25 | 98.36 ± 13.10 | -0.84 (-0.85%) | 0.435 |
| 4 | 138.78 ± 26.58 | 116.21 ± 15.08 | **-22.56 (-16.26%)** | **<0.001** |
| 5 | 147.86 ± 26.59 | 113.64 ± 19.52 | **-34.21 (-23.14%)** | **<0.001** |
| 4 and 5 | 286.63 ± 31.77 | 229.86 ± 22.40 | -56.77 (-19.81%) | **<0.001** |

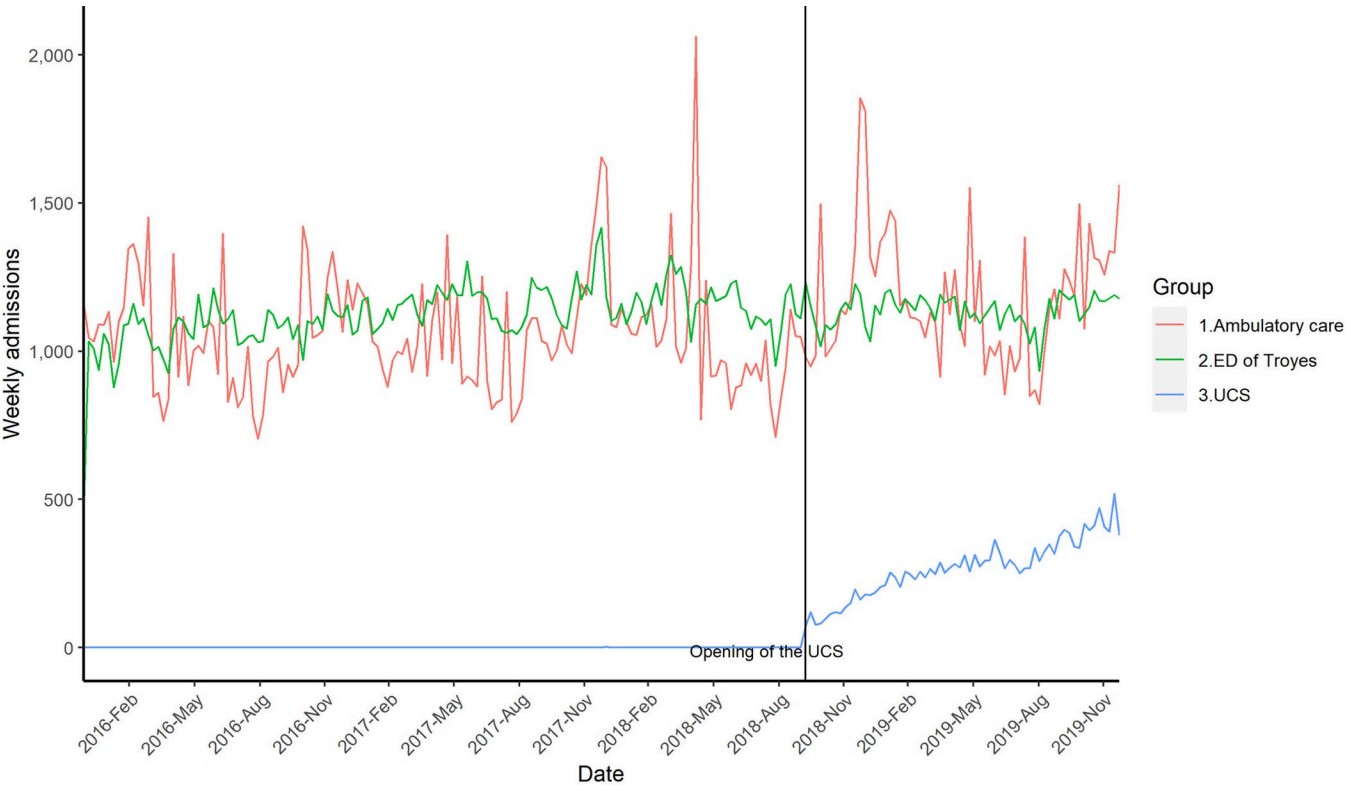

**Fig 4. Trends in unscheduled care in the Aube Region of France from December 2015 to December 2019.** The numbers of weekly admissions is represented using lines for each type of unscheduled care in the area surrounding the ED of Troyes. Ambulatory care is all the unscheduled consultations performed for the most part by general practitioners and nurses in the Aube territory via the SOS Médecin association and the Permanence Des Soins Ambulatoires (PDSA) registered in the Système Nationale des Données de Santé (SNDS, the National Health Information database).

Our study applied a novel method of analyzing multimorbidities by clustering the general population of ED patients. To the best of our knowledge and based on the review of Padros-Torres et al focused on complex patient and chronic diseases [21–25], this is the first time that an analysis of multimorbidity patterns has been used to cluster healthcare visits using a temporal-based analysis within a specific population. With this novel method [34], we extended the use of these methods to a general ED population with a low multimorbidity context and with a comprehensive clustering system that was able to classify the vast majority of ED visits (92.93%). Other classifications based on the acuity or urgency of each patient such as the CIMU scale, CCMU scale or PS classification could have directly been used, especially since UCS targets low-acuity patients that should not require hospitalization or vital care that was shown with patients of CIMU levels 4 and 5. However, this classification does not give any clinical information on the patient's health problem and therefore has a limited potential for interpretation. As such, the conjunction of the CIMU scale and the multimorbidity classification developed here seems a great tool to analyze the ED patient population and their attendance.

Previous studies investigating the impact of UCS have reported substantial reductions in "avoidable" ED visits as well as cost reductions [11–15]. Using a questionnaire, Moe et al. measured the impact of an after-hours clinic in terms of avoided ED visits and cost savings. This study found that 36.8% of patients going to the after-hours clinic would otherwise have gone to the ED if the clinic had not been available [11]. Patwardhan et al. [13] found that among more than 2.6 million encounters in the absence of convenient care clinics (CCCs), 4.5% of all

**Table 4. Logistic regression model showing the factors associated with the probability of belonging to decreasing clusters.**

| Variable | OR | 95%CI |
|---|---|---|
| After period (Reference: Before) | 0.83 | [0.80,0.85] |
| Readmission within one week | 0.58 | [0.55,0.61] |
| Readmission between one week and 30 days | 0.60 | [0.57,0.64] |
| Gender (Female) | 0.72 | [0.70,0.74] |
| PS1 status | 1.14 | [1.09,1.18] |
| Age category (Reference: 25 to 50 years) | | |
| Short circuit through ED | 2.26 | [2.18,2.35] |
| *0 to 10 years* | 0.71 | [0.68,0.74] |
| *25 to 50 years* | - | - |
| *10 to 25 years* | 1.26 | [1.21,1.31] |
| *50 to 75 years* | 1.32 | [1.27,1.37] |
| *75 years and over* | 1.82 | [1.74,1.90] |
| Drug prescriptions (per drugs administrated) | 0.97 | [0.96,0.97] |
| Number of biological examinations (per additional exam) | 0.68 | [0.66,0.69] |
| Number of radiological examinations (per additional exam) | 1.36 | [1.33,1.39] |

Note: All parameters were tested with a p<0.001 at Wald test (z-test), AUC = 0.68. PS1: Patient State class 1; ED: Emergency Department.

Hosmer-Lemeshow test: $X^2$ = 658.16, p<1e-16.

patients would have gone to the ED for weekend and after-hours visits and 3.15% for weekday consultations during office hours while 29.39% and 27.34% would have gone to an urgent care center, respectively, when investigating the use of CCCs operating outside of office hours.

In our study, the observed rate of 30.58% was similar to that found in literature. In a before-and-after study of monthly ED visit frequency after implementation of an after-hours clinic, Jones et al. reported a mean of 49.28 fewer semi-urgent patient visits to the ED per month [15]. Moreover, with an average reduction of 57.95 visits per week, which corresponded to a reduction of 13.59% in the 7 clusters with decreases and a reduction of 4.96% of all ED visits, our findings are in the same range as those in previous reports. These results can be explained by the large proportion of "avoidable" visits among those going to the ED and patients are often characterized by low acuity and high vulnerability [5, 40, 41]. Redirecting them towards other healthcare facilities yields cost savings.

The major obstacles to greater success of UCS are likely coming from the behavior of patients, some of whom may go to the ED because they are not aware of the alternatives or because there is no financial incentive for them to try less costly care options first [4, 42–45]. In this regard, public health campaigns or the introduction of co-payment fees for ED visits can be helpful in limiting "avoidable" ED visitations. Also, the choice of patients is influenced by their perception of quality of care in each of the facilities. In the present context, with the bad reputation of the ED for non-urgent stay, a patient that knows the existence of the UCS alternatives and knows that it is more suitable for its emergent health problem would choose to go to these. Another obstacle, that was not present here, was the potential difficulty for patients to quickly access the UCS due to their location. This obstacle should be addressed before the implementation of the solution since it cannot be fixed afterwards [46]. Fortunately, the four obstacles being knowledge, cost, distance, and reputation presented here can all be prevented with good planning, information campaign and political reform. Progressively eliminating these obstacles should yield a better flow redirection toward UCS. However, unscheduled consultations remain

a weak link in the delivery of programmed healthcare and are also more difficult to manage. The creation of healthcare facilities specifically designed for unscheduled care relies on public demand for such services which can result in an increase in overall costs and the reappearance of saturation and crowding problems regarding ambulatory unscheduled care. To put it differently, the availability of emergent care solutions will make some patients prefer this solution even if it is not necessary. Since improving access to primary care seems to be insufficient to alleviate the demand for total unscheduled care [47, 48], other solutions remain to be found to limit social vulnerability and improve the general health of the public [43, 49].

This study had some limitations. Firstly, there was a potential for selection bias in the population of Troyes. Nevertheless, this bias was minimized by previous reports in various literature showing that the population could cohesively be representative of the national population [32]. Secondly, in this study there was no control group that was unaffected by the opening of a UCS facility. The conditions of access to healthcare data from other French regions and the specificity of the local context precluded us from using a design that might limit this bias. Indeed, finding a control that corresponded to our facility that would not have been affected by the opening of new UCS was impossible. Thus, separating the real causal impact of opening new structures from other global, seasonal and/or random variations presented an overly arduous task. Another approach that would have potentially allowed for a controlled design would have been assessing and modelling the change in the probability of patient attendance in a given territory, however this would have required access to national databases [50]. Thirdly, the choice of methods for clustering and the differential analyses might have been limited our findings since there were other state-of-the-art approaches that are frequently used for this type of study.

As stated in the methods section, other clustering approaches exist, however, they could not have given a meaningful clinical repartition that would allow us to infer any possible link with the impact of new UCS facilities. As illustrated in S1 Fig, the use of ICD10 chapters to classify ED visits could have also been used and may have given some indication of temporal trends, especially with the large decrease of visits classified under Chapter XIX (S00T88): *Injury, poisoning and certain other consequences of external causes*. Albeit clinical, this classification does not result from a clustering process like the one identified here where the groups of ICD10 blocks held a statistical meaning of similarity; meaning a readmission from a patient may be more likely to result in a diagnostic from the same cluster as that of their initial visit. As such, clustering may give a more appropriate clinical and statistical representation of health problems with a better distribution of visit volumes between clusters.

Other designs of clustered differential analysis could have been employed by using either differences-in-differences [51] or regression discontinuity [52]. The trend analysis method used in this study was chosen because the only control data were the admissions during the before period. Indeed, the opening of the UCS can be seen as an intervention but its effect is hard to control since an experimental setting would have required a national scientific and economic effort. Our observational design was empirically built such that if there is constant yearly pattern, no trend would be seen. It requires taking several years to have full periods and not be biased by local variations during a year. The evaluation period was limited to six months to impose a strong decreasing effect and to limit the potential bias. Furthermore, the additive Poisson regression model was chosen over a multiplicative one due to the limited amount of variation and the interpretability of the additive result that is more congruent with the selected visit per week outcome than multiplicative coefficients [37]. Clearly, the approach in this study relied on the quality of disease clusters and proposed a pertinent and pragmatic evaluation of the nature of the changes in ED attendance with the logistic model serving as a confirmatory tool of these changes. As a future statistical contribution, a full comparison with the results and the methods of different approaches could deepen the understanding of the impact effect.

Lastly, this study did not look at all the saturation factors of ED when elements other than arrival admissions such as the mean length of stays, mean number of patients waiting, and the bottleneck situation probabilities inside the ED and for hospitalization were present. Future research on this matter should be taken into account since one of the main overcrowding factors is directly due to institutional reasons of the operational behavior of hospitals with the boarding time of patients to hospitalize [53].

## Conclusion

This study presented an original and reliable method for assessing changes in patient flow through different healthcare delivery services that can be extrapolated to several contexts where only the existence of a system for labeling diagnoses so that their co-occurrence can be measured in the patients' records is required. Moreover, our findings showed that it can be efficacious when patients are characterized according to hierarchical codes such as the ICD10 diagnostic system and can also be successfully applied to study temporal trends in relation to a specific event even in the absence of a control group. With this methodological approach current public health policy considerations may be conformed and reorganizing the relationships between primary care and hospital EDs with a goal of improving the functioning of EDs by boosting the offer of unscheduled and/or non-urgent care access may be aimed. The implementation of new UCS was accompanied by a decrease in ED visits among multiple patient profiles and accords with the type of patients cared for by the new services. This new distribution of healthcare delivery therefore may have a concrete impact and help optimize patient flow arrival to the different types of care facilities.

## Supporting information

**S1 Table. Diagnostic content and quality indicators of the 14 clusters.**
(DOCX)

**S2 Table. Labels of the ICD10 blocks with content of each block from the ICD10 chapter classification structure.**
(DOCX)

**S3 Table. Coefficients results Poisson regression model without global trend of emergency department (ED) visits counts per day from 2016 to 2019.**
(DOCX)

**S4 Table. Coefficient result Poisson regression model with global trend of emergency department (ED) visits counts per day from 2016 to 2019.**
(DOCX)

**S1 Fig. Change in weekly admissions to illustrate the trends for the 16-cluster structure and the ICD10 chapter classification structure from 2017 to 2019.**
(DOCX)

**S1 File.**
(DOCX)

## Acknowledgments

The authors would like to thank Fiona Ecarnot for her contribution in translating the study and AcaciaTools for their medical writing and reviewing services.

## Author Contributions

**Conceptualization:** Adrien Wartelle, Farah Mourad-Chehade, Farouk Yalaoui.

**Data curation:** Jan Chrusciel.

**Formal analysis:** Jan Chrusciel.

**Supervision:** David Laplanche, Stéphane Sanchez.

**Validation:** Hélène Questiaux, Thomas Monneret, Jan Chrusciel, Stéphane Sanchez.

**Writing – original draft:** Hélène Questiaux, Thomas Monneret, Ghislain Soliveau, David Laplanche, Stéphane Sanchez.

**Writing – review & editing:** Hélène Questiaux, Thomas Monneret, Ghislain Soliveau, Jan Chrusciel, Antoine Duclos, David Laplanche, Stéphane Sanchez.

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
