## [Decision Letter · Decision Letter 0]

2 Aug 2021

PONE-D-21-18947

Multimorbidity Clustering of the Emergency Department Patient Flow: Impact Analysis of New Unscheduled Care Clinics

PLOS ONE

Dear Dr. Sanchez,

Thank you for submitting your manuscript to PLOS ONE. After careful consideration, we feel that it has merit but does not fully meet PLOS ONE’s publication criteria as it currently stands. Therefore, we invite you to submit a revised version of the manuscript that addresses the points raised during the review process.

We look forward to receiving your revised manuscript.

Kind regards,

Yong-Hong Kuo

Academic Editor

PLOS ONE

Journal Requirements:

2. You indicated that ethical approval was not necessary for your study since this was as retrospective study based on fully anonymized data as such ethics approval is not required under the French Law. Please clearly clarify this within the ethics statement of the manuscript text and the online submission form, ensuring that you have specified that all data was fully anonymised prior to access

Additional Editor Comments (if provided):

The submission has been reviewed by two experts in the area. Both of them believe that this work has merit and is interesting to the reader. Both referees have provided constructive suggestions to the authors to improved the work. The authors shall consider them in the revision.

Reviewers' comments:

Reviewer's Responses to Questions

**Comments to the Author**

1. Is the manuscript technically sound, and do the data support the conclusions?

Reviewer #1: Yes

Reviewer #2: Yes

2. Has the statistical analysis been performed appropriately and rigorously? 

Reviewer #1: Yes

Reviewer #2: Yes

3. Have the authors made all data underlying the findings in their manuscript fully available?

Reviewer #1: Yes

Reviewer #2: Yes

4. Is the manuscript presented in an intelligible fashion and written in standard English?

Reviewer #1: Yes

Reviewer #2: Yes

5. Review Comments to the Author

Reviewer #1: I believe that it is a useful study for the emergency room overcrowding, which has become a global problem. In the study of the authors, it was observed that there was no ethical defect or a situation where there could be a conflict of interest.

Reviewer #2: Dear editor in chief, dear authors

I thank the Editor for the opportunity to review this manuscript which investigates the impact of the opening of unscheduled care structures on the activity of an emergency service. The authors propose an analysis based on multmorbidity clustering of the ED patients. This analysis is very interesting. However, the article requires revisions before it can be considered for publication.

Major points:

Method

Line 143: Between October and November 2018 the opening of two new UCS facilities offering services akin to convenient care clinics changed the organization of emergency care delivery in the city.

It is important to locate the UCS in relation to the hospital to know if access is easy, which would be an argument for assuming that patients who go to the hospital are those who go to these facilities

In addition, it is important to describe the health capacity of the city: number of clinics and other facilities where the hospital patients could have gone.

Line 150: The 2-year study period was chosen to account for the yearly seasonal variations

However, in line 197, it appears that the study is done 18 months before and 6 months after. What is the justification for not taking a year after opening to have the variation over a full year?

I am not an expert in statistics, nevertheless an time series analysis could be used considering that opening UCS is an intervention.

Results

Page 9, Line 216: Who enters the ICD10 diagnosis codes? Is it emergency physicians only or can others such as residents or non-medical staff do the entry? How is the reliability of code entry verified in the hospital? This data should be clarified to better understand the analysis of evolution of ED visits

Discussion

Page 11, Line 275: We understand the decrease in traumatological pathologies with the opening of such units, but how can we explain the increase in infectious and chronic pathologies? Hypotheses could be evoked to develop the discussion. In any case, it shows the change of behavior in the society and the way of treating oneself.

In addition, it is important to describe the health capacity of the city: the number of clinics and other facilities to which the hospital patients could have gone. For example, if a private hospital opened, this may also influence patient flow. Is it easy to get to the hospital and the UCS? These are also factors that can influence the choice of going to a particular facility, beyond the economic factor.

Studies exist on the profile of patients consulting ED for non-urgent pathologies and on the choice of patients to come to ED rather than to use such facilities. it would be desirable to discuss the factors that could influence patients to go to ED or to these facilities

This element could be discussed

Finally, an analysis of flows based on patient severity could be interesting, for example, based on a classification of patients according to triage scales (Manchester triage scale, Canadian triage and acuity scale, The Emergency Severity Index, CIMU in France…). Are the patients who used to come to the emergency room and who use these new structures the least serious? The choice of ICD10 rather than these scales should be discussed?

Minor points

Methods

Line 157 : triage path

For those who are not familiar with the organization of EDs, it is difficult to understand the triage process. We can imagine that the scale used is a validated scale (like the Canadian triage and acuity scale, CIMU in France, or other...)

From which software are the data of the patients consulting the emergency room extracted?

Results

Adding the estimated size of effect and its precision communicates what range of outcomes would be probable if the trail were repeated multiple times. It would be interesting to have the confidence interval (for example line 261)

The authors should probably limit the number of decimal places for the p-value, and I suggest noting <0.0001 for values below this threshold

Tables:

Table 2 should include the IC95% to better analyze the differences obtained.

Table 3: legend should be added (PS1, ED...)

Figures:

Figure 3: the scale of the y-axis could be changed to make the box plot more readable

Once again, thank you for the opportunity to review this article.

Best regards

DAG

6. PLOS authors have the option to publish the peer review history of their article (what does this mean?). If published, this will include your full peer review and any attached files.

Reviewer #1: No

Reviewer #2: **Yes: **Daniel Aiham GHAZALI

---

## [Author Response · Author response to Decision Letter 0]

24 Sep 2021

aResponse to Journal comments

Journal Requirements: When submitting your revision, we need you to address these additional requirements.

 Please ensure that your manuscript meets PLOS ONE's style requirements, including those for file naming. The PLOS ONE style templates can be found at:

Authors Response:

After carefully reading the templates as well as the author guidelines, the manuscript has been entirely adjusted to meet the style requirements such as file naming, figure and table naming and reference citations. 

 You indicated that ethical approval was not necessary for your study since this was as retrospective study based on fully anonymized data as such ethics approval is not required under the French Law. Please clearly clarify this within the ethics statement of the manuscript text and the online submission form, ensuring that you have specified that all data was fully anonymized prior to access.

Authors Response:

Thank you for this comment. The revised manuscript has been modified. The ethics statement was placed in the method section of the manuscript clearly stating that:

“Since the study was wholly observational and only used anonymized data (patient names were not recorded), neither ethics approval nor a specific written informed consent from participants were required under French law as a retrospective database study”. 

Furthermore, the French Public Health Code was cited to further justify the absence of a written consent and of an ethical committee. The study was conducted with all medical information related legislation and was declared to the National Registry of Health research body under N°1113130319.

 Thank you for stating the following financial disclosure: "The funders had no role in study design, data collection and analysis, decision to publish, or preparation of the manuscript."

 Please clarify the sources of funding (financial or material support) for your study. List the grants or organizations that supported your study, including funding received from your institution. 

 State what role the funders took in the study. If the funders had no role in your study, please state: “The funders had no role in study design, data collection and analysis, decision to publish, or preparation of the manuscript.”

 If any authors received a salary from any of your funders, please state which authors and which funders.

 If you did not receive any funding for this study, please state: “The authors received no specific funding for this work.”

Authors Response:

Thank you for this comment. There was no specific funding for this work, neither financial or material support. There were no grants or organizations that supported the study including funding from our institution. There were no funders to take any role in study design, data collection and analysis, decision to publish, or preparation of the manuscript. The authors received no specific funding for this work. This will be added to the cover letter. 

Additional Editor Comments (if provided):

The submission has been reviewed by two experts in the area. Both of them believe that this work has merit and is interesting to the reader. Both referees have provided constructive suggestions to the authors to improve the work. The authors shall consider them in the revision.

Authors Response:

We thank these reviewers for their remarks and have considered them in the revised manuscript. Indeed, Reviewer 2 had constructive suggestions with minor and major points. However, we did not find any specific remarks from the Reviewer 1 in your mail, such that we understood that this work was ready for publication for them.

Response to Reviewer 1 comments

Reviewer #1: I believe that it is a useful study for the emergency room overcrowding, which has become a global problem. In the study of the authors, it was observed that there was no ethical defect or a situation where there could be a conflict of interest.

Authors Response:

We sincerely thank Reviewer 1 for reading our paper and providing their appreciation towards it.

 

Response to Reviewer 2 comments

Reviewer #2: Dear editor in chief, dear authors

I thank the Editor for the opportunity to review this manuscript which investigates the impact of the opening of unscheduled care structures on the activity of an emergency service. The authors propose an analysis based on multmorbidity clustering of the ED patients. This analysis is very interesting. However, the article requires revisions before it can be considered for publication.

Major points:

Method

 Line 143: Between October and November 2018 the opening of two new UCS facilities offering services akin to convenient care clinics changed the organization of emergency care delivery in the city.

It is important to locate the UCS in relation to the hospital to know if access is easy, which would be an argument for assuming that patients who go to the hospital are those who go to these facilities

In addition, it is important to describe the health capacity of the city: number of clinics and other facilities where the hospital patients could have gone.

Authors Response:

Thank you for this comment. We recognized that the description of the health capacity of Troyes and its surrounding was crucial to assess the results and so we clarified the situation in the method section without exacerbating too much on the amount of text (L89):

“These two new structures are the Aubois Establisment of Immediate Care situated in the north of the city at about 4 km from the hospital and the Polyclinic of Montier La Celle at about 1.5km. They are as easy to access as the ED for the surrounding population and thus their localization does not cause any flow constraint. The three structures represent the main unscheduled services in the city and in the department with only one other ED in the Aube department in Romilly sur-Seine accounting for approximately 20,000 ED visits in 2017. They are completed by the Ambulatory unscheduled care, performed for the most part by general practitioners and nurses in the Aube territory via the SOS Médecin association and the Permanence Des Soins Ambulatoires (PDSA). The Hospital of Troyes entails almost all the hospitalization health capacity of the city and is a necessary passage point for all inpatients requiring emergent care. More information about the health capacity of the city can be found on the FINESS file freely available on internet, or through the hospital website”.

2) Line 150: The 2-year study period was chosen to account for the yearly seasonal variations

However, in line 197, it appears that the study is done 18 months before and 6 months after. What is the justification for not taking a year after opening to have the variation over a full year?

I am not an expert in statistics, nevertheless an time series analysis could be used considering that opening UCS is an intervention.

Authors Response:

Thank you very much for such an important comment. The revised manuscript has been modified regarding the study design in the Methods, Results and Discussion sections as a result with the following text: 

 Methods section (L166):

“In addition to the control for typical seasonal trends, the duration of the after periods with its 1 to 3 ratio, was designed such that we could quantify the medium-term impact where the variations in ED visits had more chance to be directly related to the opening of the new structures than in the long-term case where other complex effects and trends, such as the COVID-19 pandemic, can take place and mitigate the result significance through confounding.

To adjust this for seasonal confounders and obtain confidence interval, the differential analysis was complemented using a time series analysis of the ED visits from the year 2016 to 2019. Each cluster daily visit count V_ED (t) (t: day index) over the 4-year period was analyzed with a Poisson distributed linear regression with the retained model inspired by the literature [36,37]:

V_ED (t)=a_0+m(t)^' a_1+d(t)^' a_2+p(t)^' a_3

where a_0 is the intercept, a_1 to a_3 the regressed linear vector coefficients on the month boolean indicator vector m(t), the day of week boolean indicator vector d(t) and the period boolean indicator vector p(t) with 2 before periods 2016-01-01 to 2017-05-01 and 2017-05-01 to 2018-10-08 (taken as reference) and 2 after periods 2018-10-08 to 2019-04-28 and 2019-04-28 to 2019-12-31. Another model was also applied to consider a global trend effect a_4 over t.”

 Result section (L23 bis):

“The Poisson regression model coefficients results available in S4 Table showed consistent but smaller results with an overall decrease of 39.83 visits per week (95% CI [24.49,55.13]) and a specific decrease in the seven clusters of 39.17 visits (95% CI [30.25, 48.06]). The more pronounced overall decrease could be linked to the accounting of seasonal effects in Cluster 3 and Cluster 4 where the decrease of 4.09 (95% CI [-1.50,9.63]) and the increase of 3.30 (95% CI [-2.19,8.83]) were much smaller and not significant. The trends of the initial decreasing Clusters 2, 6, 7, 9, 11, 12 and 15 were 8.40 (95% CI [3.03,13.74]), 8.39 (95% CI [4.76,11.98]), 6.21 (95% CI [2.71,9.67]) 8.00 (95% CI [11.78,17.17]), 3.41 (95% CI [0.84,5.94]), 2.51 (95% CI [0.03,5.01]) and 1.70 (95% CI [0.41,3.78]) respectively. They tended to be lower than the empirical observations and Cluster 12 and 15 were not significant anymore. The Poisson regression model with global trend described in S5 Table suggest a more pronounced overall decrease of 85.37 (95% CI [65.58, 105.16]) and specific decrease in Clusters 2, 6, 7, 9, 11, 12 and 15 of 48.99 (95% CI [37.33,60.63]). Using this model that described a more global effect, we would have selected Cluster 1, 2, 3, 6, 7, 9 and 11 that were decreasing significantly.”

 Discussion section (L104 bis and L187 bis):

“The Poisson regression model with global trend described in S5 Table also suggests that the UCS disrupted the linear trend of ED visits growth that were taking place before 2019 even though the decrease observed is still inferior and the estimation and interpretation of a global linear trend is questionable.”

And,

“Indeed, the opening of the UCS can be seen as an intervention but its effect is hard to control since an experimental setting would have required a national scientific and economic effort. Our observational design was empirically built such that if there is constant yearly pattern, no trend would be seen. It requires taking several years to have full periods and not be biased by local variations during a year. The evaluation period was limited to six months to impose a strong decreasing effect and to limit the potential bias.”

Results

3) Page 9, Line 216: Who enters the ICD10 diagnosis codes? Is it emergency physicians only or can others such as residents or non-medical staff do the entry? How is the reliability of code entry verified in the hospital? This data should be clarified to better understand the analysis of evolution of ED visits

Authors Response:

Thank you for this comment. We clarified this point a little bit further in the methods section with the following text added to the revised manuscript (L126):

“….this assessment is made at the end of each stay by the current emergency physicians and resident medical staff with evaluation of all the information given during a patient stay in the ED and is further verified by the medical information team for coherence”

Discussion

4) Page 11, Line 275: We understand the decrease in traumatological pathologies with the opening of such units, but how can we explain the increase in infectious and chronic pathologies? Hypotheses could be evoked to develop the discussion. In any case, it shows the change of behavior in the society and the way of treating oneself.

Authors Response:

Thank you for this comment. To our knowledge, this temporary increase is due to seasonal variations of epidemics since there is more time spent in an epidemic timeframe in the after period. We already pointed out this explanation in the discussion section (L97 bis) of the revised manuscript which is now confirmed with the new Poisson regression model:

“As illustrated in S3 Figure and shown by the Poisson regression model in S4 Table, the temporal variations in the patient profiles of the two clusters are only due to seasonal variations notably during the winter periods of epidemics and crowding of the ED that are more present in the after period, rather than a tangible trend related to the opening of UCS”.

5) In addition, it is important to describe the health capacity of the city: the number of clinics and other facilities to which the hospital patients could have gone. For example, if a private hospital opened, this may also influence patient flow. Is it easy to get to the hospital and the UCS? These are also factors that can influence the choice of going to a particular facility, beyond the economic factor.

Authors Response:

Thank you for this comment. This point has already been addressed with Remark 1 but we added the two following sentences in the discussion section of the revised manuscript (L146 bis) to point out this obstacle. 

“Another obstacle, that was not present here, was the potential difficulty for patients to quickly access the UCS due to their location. This obstacle should be addressed before the implementation of the solution since it cannot be fixed afterwards [46].”

6) Studies exist on the profile of patients consulting ED for non-urgent pathologies and on the choice of patients to come to ED rather than to use such facilities. it would be desirable to discuss the factors that could influence patients to go to ED or to these facilities

This element could be discussed

Authors Response:

Thank you for this comment. We think that the four obstacles addressed now in the revised discussion section (L143 bis) are the main factors regarding the choice of emergent care facility:

“Also, the choice of patients is influenced by their perception of quality of care in each of the facilities. In the present context, with the bad reputation of the ED for non-urgent stay, a patient that knows the existence of the UCS alternatives and knows that it is more suitable for its emergent health problem would go to these. Another obstacle, that was not present here, is the potential difficulty to access quickly to the UCS due to their location. This obstacle should be addressed before the implementation of the solution as it cannot be fixed afterwards [46]. Fortunately, the four obstacles being knowledge, cost, distance, and reputation presented here can all be prevented with good planning, information campaign and political reform. Progressively eliminating these obstacles should wield a better flow redirection toward UCS.”

Though it should be noted that before choosing a facility, there is the choice to resort to unscheduled care that is directly influence by its availability, even if unnecessary.

7) Finally, an analysis of flows based on patient severity could be interesting, for example, based on a classification of patients according to triage scales (Manchester triage scale, Canadian triage and acuity scale, The Emergency Severity Index, CIMU in France…). Are the patients who used to come to the emergency room and who use these new structures the least serious? The choice of ICD10 rather than these scales should be discussed?

Authors Response:

Thank you for this comment. After some investigation, we were able to extract the CIMU classifications of patient’s stay and use it for further analysis in conjunction with the clustering. The revised manuscript has been modified with the following text in the following sections:

 In the Method section (L159):

“The clusters showing a significant decrease were then grouped and decomposed using the CIMU scale to perform another differential analysis and show which acuity of patient was the most impacted”

 In the Results section with new table 3 (L22 bis):

“As illustrated in Table 3, further analysis using the CIMU scale show that only CIMU levels 4 and 5 patients are responsible for a significant decrease of 56.77 visits per week.”

 In the Discussion section (L92 bis):

“The decrease predominantly involved patient profiles linked to trauma and trauma-related symptoms that are low-acuity, meaning they are strictly related to CIMU levels 4 and 5 with patients going predominantly to short circuit”

 In the Discussion section (L114 bis):

“Other classifications based on the acuity or urgency of each patient such as the CIMU scale, CCMU scale or PS classification could have directly been used, especially since UCS targets low-acuity patients that should not require hospitalization or vital care that was shown with patients of CIMU levels 4 and 5. However, this classification does not give any clinical information on the patient’s health problem and therefore has a limited potential for interpretation. As such, the conjunction of the CIMU scale and the multimorbidity classification developed here seems a great tool to analyze the ED patient population and their attendance.”

Minor points

Methods

8) Line 157: triage path

For those who are not familiar with the organization of EDs, it is difficult to understand the triage process. We can imagine that the scale used is a validated scale (like the Canadian triage and acuity scale, CIMU in France, or other...)

Authors Response:

Thank you for this comment. The triage process is indeed based on CIMU scale and is clarified in the Method section of the revised manuscript (L115):

“This triage process is done by a triage nurse which evaluates the urgency of the patient through the CIMU scale (Nurse Classification of ED Patients) [33]. This scale has five levels (1, 2, 3, 4 and 5) corresponding in the current context to a necessity of an immediate intervention, of an intervention below 20mins, 60mins, 120 mins and 240 mins respectively. The triage evaluation is directly related to the decision sent to the patient to short circuit, long circuit, or vital emergency room for reanimation. During the study period, levels 1, 2, 3, 4 and 5 patients were sent in long circuit or vital emergency for 92.03% of stays (63.2% being vital), 85.55% (3.5% vital), 62.30% (0.54% vital), 15,26% (0.13% vital) and 6.3% (0.08% vital) respectively.”

9) From which software are the data of the patients consulting the emergency room extracted?

Authors Response:

Our paper did not include any software sub information as such, we have added a method sub-section in the revised manuscript with sufficient information as follows (L209):

“The data used in this study was extracted using from the software’s database of the ED of Troyes, named RESURGENCES, and using SQL request through the SQL developer software. This software is directly used by the physicians and residents of the ED to register patients’ data during their stay. The data was analyzed using R 4.0.2 in the RStudio1.2.5042 environment with the packages Tidyverse 1.3.0, Lubridate 1.7.9, stats 4.0.2, and Treemap 2.4.2.”

Results

10) Adding the estimated size of effect and its precision communicates what range of outcomes would be probable if the trail were repeated multiple times. It would be interesting to have the confidence interval (for example line 261)

Authors Response:

Thank you for this comment. The Poisson regression model allows us to precise a size effect with a confidence interval:

“The Poisson regression model coefficients results available in S4 Table shows consistent but smaller results with an overall decrease of 39.83 visits per week (95% CI [24.49,55.13]) and a specific decrease in the 7 clusters of 39.17 visits (95% CI [30.25, 48.06]).”

11) The authors should probably limit the number of decimal places for the p-value, and I suggest noting <0.0001 for values below this threshold

Authors Response:

Thank you for this comment. After checking the guide for authors, we limited the number of decimal places with the minimal written p-value being 0.001 (written below as p<0.001). The revised manuscript has been modified. 

Tables:

12) Table 2 should include the IC95% to better analyze the differences obtained.

Authors Response:

Thank you for this comment. These differences were empirical observations and a full Poisson regression model has been added to the revised manuscript with the CI 95%.

13) Table 3: legend should be added (PS1, ED...)

Authors Response:

Thank you for this comment. The legend has been added in the revised manuscript. 

Figures:

14) Figure 3: the scale of the y-axis could be changed to make the box plot more readable

Authors Response:

Thank you for this comment. The pad break was set to 50 to increase the readability in the revised manuscript.

Once again, thank you for the opportunity to review this article.

Best regards

DAG

Authors Response:

Thank you for this detailed and thorough review of our work. It has allowed us to dramatically improve the manuscript.

---

## [Decision Letter · Decision Letter 1]

10 Jan 2022

Multimorbidity clustering of the Emergency Department patient flow: impact analysis of new unscheduled care clinics

PONE-D-21-18947R1

Dear Dr. Sanchez,

We’re pleased to inform you that your manuscript has been judged scientifically suitable for publication and will be formally accepted for publication once it meets all outstanding technical requirements.

Kind regards,

Yong-Hong Kuo

Academic Editor

PLOS ONE

Additional Editor Comments (optional):

The referee who had concerns about the work reviewed the revision and is satisfied with the revised work. Based on the recommendations and comments from the referees of the two rounds, I recommend Accept.

Reviewers' comments:

Reviewer's Responses to Questions

**Comments to the Author**

1. If the authors have adequately addressed your comments raised in a previous round of review and you feel that this manuscript is now acceptable for publication, you may indicate that here to bypass the “Comments to the Author” section, enter your conflict of interest statement in the “Confidential to Editor” section, and submit your "Accept" recommendation.

Reviewer #2: All comments have been addressed

2. Is the manuscript technically sound, and do the data support the conclusions?

Reviewer #2: Yes

3. Has the statistical analysis been performed appropriately and rigorously? 

Reviewer #2: Yes

4. Have the authors made all data underlying the findings in their manuscript fully available?

Reviewer #2: Yes

5. Is the manuscript presented in an intelligible fashion and written in standard English?

Reviewer #2: Yes

6. Review Comments to the Author

Reviewer #2: Dear editor in chief, dear authors

I thank the Editor for the opportunity to review this revised manuscript which investigates the impact of the opening of unscheduled care structures on the activity of an emergency service.

I thank the authors for responding to all the comments, especially regarding the use of the times series analysis, considering that opening UCS is an intervention.

Best regards

DAG

7. PLOS authors have the option to publish the peer review history of their article (what does this mean?). If published, this will include your full peer review and any attached files.

Reviewer #2: **Yes: **Daniel Aiham GHAZALI

---

## [Editor Report · Acceptance letter]

21 Jan 2022

PONE-D-21-18947R1 

Multimorbidity clustering of the Emergency Department patient flow: impact analysis of new unscheduled care clinics 

Dear Dr. Sanchez:

I'm pleased to inform you that your manuscript has been deemed suitable for publication in PLOS ONE. Congratulations! Your manuscript is now with our production department. 

Kind regards, 

on behalf of

Dr. Yong-Hong Kuo 

Academic Editor

PLOS ONE